

# Recommendation algorithm based on attributed multiplex heterogeneous network

Zhisheng Yang and Jinyong Cheng

Qilu University of Technology (Shandong Academy of Sciences), Jinan, China

## ABSTRACT

In the field of deep learning, the processing of large network models on billions or even tens of billions of nodes and numerous edge types is still flawed, and the accuracy of recommendations is greatly compromised when large network embeddings are applied to recommendation systems. To solve the problem of inaccurate recommendations caused by processing deficiencies in large networks, this paper combines the attributed multiplex heterogeneous network with the attention mechanism that introduces the softsign and sigmoid function characteristics and derives a new framework SSN_GATNE-T (S represents the softsign function, SN represents the attention mechanism introduced by the Softsign function, and GATNE-T represents the transductive embeddings learning for attribute multiple heterogeneous networks). The attributed multiplex heterogeneous network can help obtain more user-item information with more attributes. No matter how many nodes and types are included in the model, our model can handle it well, and the improved attention mechanism can help annotations to obtain more useful information *via* a combination of the two. This can help to mine more potential information to improve the recommendation effect; in addition, the application of the softsign function in the fully connected layer of the model can better reduce the loss of potential user information, which can be used for accurate recommendation by the model. Using the Adam optimizer to optimize the model can not only make our model converge faster, but it is also very helpful for model tuning. The proposed framework SSN_GATNE-T was tested for two different types of datasets, Amazon and YouTube, using three evaluation indices, ROC-AUC (receiver operating characteristic-area under curve), PR-AUC (precision recall-area under curve) and F1 (F1-score), and found that SSN_GATNE-T improved on all three evaluation indices compared to the mainstream recommendation models currently in existence. This not only demonstrates that the framework can deal well with the shortcomings of obtaining accurate interaction information due to the presence of a large number of nodes and edge types of the embedding of large network models, but also demonstrates the effectiveness of addressing the shortcomings of large networks to improve recommendation performance. In addition, the model is also a good solution to the cold start problem.

Corresponding author
Jinyong Cheng, cjy@qlu.edu.cn

## INTRODUCTION

In the era of information explosion, recommendation systems (*Wu et al., 2021*) are essential to keep users engaged and satisfy personalized recommendations (*Chiu et al., 2021*). Users expect to obtain personalized content on modern e-commerce, entertainment and social media platforms, but the effectiveness of recommendations is limited by existing user-item interactions and model capacity. Therefore, some models often ignore the user-item interaction part, as it is easy to ignore the relevant information existing in the node and edge types, and the acquisition of attribute information (*Zheng, Pan & Wu, 2020*) is even more flawed. Although many models have been developed to analyse user-item interaction in the recommendation process, different models have different pros and cons regarding the user-item interaction, so the recommended effects are also different.

The nonnegligible part of the recommendation algorithm is the embedding method. Different network embedding (*Jiao et al., 2021*) methods have different recommendation effects on the recommendation algorithm and different acquisitions of node and edge-type correlation information. For user item attributes, the acquisition of information is also different, and the processing of small networks or large networks with different types of nodes and edges is also different (*da Silva et al., 2020*). As a promising direction in contemporary times, a heterogeneous information network composed of multiple types of nodes and links has emerged (*Shi et al., 2021*) (heterogeneous information network) as a general information modelling method (*Ivson et al., 2019*). HIN (Heterogeneous Information Network) is often used in recommended systems to simulate many forms of rich auxiliary data, which is inseparable from its ability to characterize various heterogeneous data very flexibly. In particular, as a relational sequence connecting pairs of objects in an HIN (*Chairatanakul, Liu & Murata, 2021*), metapaths have been widely used to extract structural features that capture the relevant semantics of recommendations (*Ding et al., 2019*). Simply put, the existing HIN-based recommendation methods can be divided into two types. The first type uses path-based semantic relevance instead of an HIN as a direct feature of recommendation relevance; the second type performs some transformations on path-based similarity for learning effective transformation features, which are then used to enhance recommendation methods of the original user or project representation (*Yang, Zhong & Woźniak, 2021*). Both of these methods are designed to improve the characterization of two-way user-item interactions, which are achieved by extracting features based on metapaths.

In the current recommendation algorithm, to better obtain the user's attributes and node information, different embedding methods, such as network embedding, heterogeneous network embedding (*Qi et al., 2021*), multichannel heterogeneous network embedding and attribute network embedding (*Xu et al., 2021*), have been produced, and each embedding method has very good development. Network embedding mainly completes graph embedding (GE) (*Manipur et al., 2021*) and graph neural networks (GNN) (*Gao et al., 2021*). Although heterogeneous networks can obtain various types of nodes or edges (*Xie et al., 2021*), they are difficult to mine due to the complex combination

of heterogeneous content and structure. In addition, research on embedded dynamic and heterogeneous data (*Hu et al., 2021*) is limited. Since there is not only a single type of proximity network between various types of nodes but also different types of proximity networks, this type of network produces a network with multiple views, which leads to the emergence of multichannel heterogeneous network embedding. Attribute network embedding (*Huang, Wang & Ma, 2021*) can find low-dimensional vector representations for the nodes in the network and preserve the original network topology and the proximity of node attributes. Although each embedding method has its advantages, these methods are used mainly in networks with a single type of node/edge. There are still shortcomings in the processing of large networks because we need to solve the problem of not only containing nodes but also fewer edge types. The network also solves the large-scale network composed of billions or even tens of billions of nodes and numerous edge types. In these large-scale networks, not only are its nodes and types numerous, but each node is associated with more than one. The two attributes are related, which causes the current existing network to be unable to solve such problems well.

In deep learning, the acquisition of information features of nodes is commonly implemented by GCN, and the one applied to low-dimensional node feature learning are MobileGCN (*Dong et al., 2021*), by using a novel affinity cod as an update in GCN, it can not only update the node features one can aggregate the features of neighbouring nodes and then obtain more node feature information and then improve the effectiveness of recommendation performance. Self-attention mechanisms are commonly used in image super-resolution processing to select image information to generate higher quality images of the Self-Attention Negative Feedback Network (SRAFBN) model (*Liu et al., 2021*), which we apply to recommender systems to improve the effectiveness of recommendation performance.

In the field of deep learning (*Russell & Reale, 2021*), large network models contain billions of nodes and edge types, but also more than one type of node and edge, and each node contains many different properties. Today's network embedding methods are mainly focused on homogeneous networks, which are characterised by a single type of node and edge. This is not enough for small networks of a wealth of different nodes and edge types, but even more so for large networks with hundreds of millions of different nodes and edge types. To solve the problem of embedding large networks of multiple nodes and multiple edge types, this paper derives a new framework SSN_GATNE-T model by combining to attribute to reuse heterogeneous networks (*Cen et al., 2019*) with an attention mechanism that introduces softsign and sigmoid function properties.

In the SSN_GATNE-T model, in order to enable the model to better handle large networks of billions or even tens of billions of different nodes and different edge types, and to better exploit the hidden information among many nodes and edge types that can potentially provide recommendations to users, we focus on combining to attribute to reuse heterogeneous networks of improved attention mechanisms, especially the improvement in attention mechanisms. In this section, we apply the characteristics of the softsign and sigmoid functions to the attention mechanism, and the purpose is to better obtain and label the associated information between different types of nodes and many edge types and

use this information to obtain more potential interests and hobbies of the user. Interactive information is used to accurately recommend interests and hobbies for the user. Applying the softsign function to the fully connected layer of the model can better reduce the potential of users mined from the information interacted by different node attributes and different edge types through attributed multiplex heterogeneous networks and improved attention mechanisms that can be used for accuracy. The loss of the recommended potential information has achieved good results in obtaining user potential information; using the Adam optimizer can make our model converge faster, and it is also very helpful for model tuning. Take the YouTube dataset as an example. It includes connections between users, sharing friends, sharing subscriptions, sharing subscribers, and sharing favourite videos. The interests of the users obtained from different interactions in the YouTube dataset are also different and should be treated differently. Our model will ignore not only the user-item interaction of these node attributes but also the potential connections between nodes and these five types of edge types. The attributed multiplex heterogeneous network can not only help us better obtain the information of each attribute but also help our more comprehensive YouTube data set with existing hidden information between the nodes and the five types of edge types, and the improved attention mechanism can mark the acquired information for mining potential information and make accurate recommendations for users. The same is true for the Amazon dataset. The only difference between the two datasets is the number of node types and edge types. Since the nodes and edge types between the two datasets have different network sizes, they also have different recommended effects.

For this paper, the main contributions are as follows.

1. The SSN_GATNE-T models addresses the shortcomings of network embedding of large network model on hundreds of millions of nodes and edge types that exist on deep learning to obtain information about the interactions between nodes and edge types, and can better handle large network models.
2. The problem of large networks acquiring node-edge type interaction defects is solved, allowing the recommendation system to acquire more potential node-edge type interaction information about recommendation of attribute reuse of heterogeneous networks and improved attention mechanism annotation, which in turn improves the effectiveness of recommendation performance.
3. Reduce the loss of potential information about users mined by the model that can be used for accurate recommendations.
4. The introduction to the Adam optimiser allows for faster convergence of the model and is also very helpful in tuning the model.

Finally, through multiple experiments of the SSN_GATNE-T model on YouTube and Amazon datasets, the improvement of its three evaluation indexes shows the effectiveness of our model for accurate recommendations. In addition, because our model does not ignore the user-item interaction information of various node attributes, it can solve the cold start problem very well.

## RELEVANT KNOWLEDGE

The network embedding method. GE (graph embedding) is an important portion of network embedding, as is a GNN (graph neural network) (*Chen et al., 2021*). DeepWalk trains a tabulation model by randomly generating a *corpus*. LINE (large-scale information network embedding) (*Jin, Zhao & Liu, 2021*) learns the representation of nodes on large-scale networks while maintaining first- and second-order approximations. Node2vec (scalable feature learning for networks) (*You et al., 2021*) designs a biased random walk program to effectively explore different communities. NetMF (network embedding as matrix factorization) (*Qiu et al., 2018*) improves the matrix factorization framework generated by the defects of DeepWalk and LINE. For GNN, GCN(graph convolutional networks) (*Huang et al., 2021*) uses a convolution operation to merge the feature representation of neighbours into the feature representation of nodes. The graph deduces an induction method that integrates node features and structural information, learning the functional representation of each node instead of embedding it directly, which helps summarize the nodes that are ignored in the training process (*Chen et al., 2021*).

The heterogeneous network embedding method. HNE (heterogeneous network embedding) (*Tang et al., 2020*) jointly considers the content and topology of the network and represents different objects in the heterogeneous network as a unified vector representation. PTE (predictive text embedding) (*Park et al., 2018*) embeds a low-dimensional space by constructing numerous heterogeneous text networks. HERec (heterogeneous information network embedding for recommendation) (*He et al., 2021*) is first transformed by the fusion function and then integrated into the extended matrix factorization model (MF).

The attribute network embedding method. TADW (text-associated deep walk) (*Berahmand, Nasiri & Li, 2021*) combines text features with network representation (*Li et al., 2021*). LANE (label-informed attributed network embedding) (*Huang, Li & Hu, 2017*) integrates label information in attribute network embedding. ANRL (Attributed network representation learning) (*Zhang et al., 2019*) is an attribute network embedding method that has a good effect on attribute information.

Multiplex heterogeneous network embedding methods. In large network models, they usually contain different types of correlation information about/on different nodes, which then gives rise to multiplex network embedding, currently commonly used are PMNE (principled multilayer network embedding), MVE (multiplex network embedding), MNE (multiplex network embedding), Mvn2vec (multi-view network embedding). PMNE proposed three methods of projecting multiplex networks of continuous vector spaces (*Pourhabibi et al., 2021*). MVE (*Xie et al., 2021*) embeds networks of multiple views of different edge types of information about nodes into a single collaborative embedding by using an attention mechanism. MNE (*Pio-Lopez et al., 2021*), on the other hand, uses a common embedding method of each node and several additional edge-types embedding methods of joint learning through a unified network embedding model. Mvn2vec (*Zhong et al., 2020*) investigates the feasibility of representing the semantics of different edges

individually in different views and then modelling, preserving and collaborating to achieve the embedding effect.

GANTE-T (*Cen et al., 2019*) and GATNE-I (*Cen et al., 2019*). The GATNE-T models aggregates information about the interaction of neighbouring nodes of different edge types of the study node, and then generates different vector representations of the nodes for each edge type to handle large networks containing hundreds of millions of nodes. The GATNE-I model is proposed to compensate for the inability of the GANTE-T model to perform inductive learning, and it is also can handle invisible node types.

FAME and FAME (*Liu et al., 2020*). The FAME and FAMEm models are fast attributing multi-way heterogeneous network embedding models for large network data containing excessive amounts of data. This model automatically preserves the attributes of nodes and the interaction information about/on different edge types at embedding time by efficiently mapping different forms of cells into the same latent space.

The method of this article. The method proposed in this article is SSN_GATNE-T, which uses base embedding and edge embedding to find different types of interactions, which can better obtain the associated information between each node and different types and can also better mark us as users. Accurately recommended associated information. Regardless of whether the node type and edge type are single or numerous, it owns the nodes. For the YouTube and Amazon datasets, the attribute characteristics of different nodes and the associated information existing between each node and different edge types are obtained through the attribute multiplexing heterogeneous network, and then the self-attention mechanism is improved to mark and obtain the information. The attribute relationship between the node and the edge type is required by the user for accurate recommendation, and then the recommendation is made through the similarity, and if the node has no feature, it is automatically generated. There are many recommended methods currently available. Compare our SSN_GATNE-T model with representative methods such as DeepWalk, metapath2vec, PMNE(c) (principled multilayer network embedding) (*He et al., 2021*) MVE (multiplex network embedding) (*Berahmand, Nasiri & Li, 2021*) GANTE-T (*Cen et al., 2019*) GATNE-I (*Cen et al., 2019*) FAME and FAME (*Liu et al., 2020*).

## MODEL

### Model architecture

Network $G = (V, \xi, A)$, $\xi = U_{r \in R} \, \xi_r$ is an attributed multiplex heterogeneous network, in which edges with edge types $r \in R$ and $|R| \geq 1$ form $\xi_r$ together. We separate the network of each edge type or consider $r \in R$ as $G = (V, \xi_r, A)$. The Attributed Multiplex HEterogeneous Network (AMHEN) describes the model architecture diagram of users and projects. Model structures with different types of nodes and different numbers of edge types are also different. For networks with more edge types and nodes, the processing results of the model are better; as the edge type category increases, the number of nodes increases, and the recommendation effect of the model enhances. Of course, the more node types and edge types there are, the more complex the AMHEN model is. The more hidden information you choose, the wider the recommended directions for

users, and the more accurate the recommendations will be. In the SSN_GATNE-T model, more than 15% of the node pairs in the two datasets we use will have more than one type of edge, and users may have multiple interaction information about the items. Our model accomplishes embedding of node as well as edge types through Base Embedding and Edge Embedding, where Base Embedding is accomplished by sharing between different types of edges contained in different nodes, and Edge Embedding is calculated by aggregating information about their neighbouring nodes through an attention mechanism. The base embedding is done by sharing between different types of edges contained in different nodes, while the edge embedding is computed by aggregating the relevant information about its neighbouring nodes through an attention mechanism. The two embedding methods are used to provide information on the interaction between the two types of nodes, user and project, and between the different edge types, and the nodes contain both user and project information. In the Youtube dataset, user input information includes gender, age and address, while item input information includes movie title and movie genre.

In order to better to describe the effectiveness of the SSN_GATNE-T model for large web embeddings and the effectiveness of recommendation performance, the model architecture of users and videos of the Youtube dataset is described in Fig. 1, which contains two node types and five edge types. The two node types are users and items with different attributes, with user inputs including information such as gender, age and address, and item inputs including information such as movie name and movie genre. The aim at combining the attribute reuse heterogeneous network model with the improved attention mechanism is to make better use of the five edge types, *i.e.* user-item interactions. The purpose of introducing the improved attention mechanism is to more explicitly labelling the access to useful information, to more fully grasp the connections between users, and to make recommendations between users who are closely connected, as the more similarities they have in common, the more similar their interests will be, and thus the better the recommendations will be. Our model SSN_GATNE-T uses mainly the interactive connection between the attributes of different nodes, determines the potential hobbies of the user and then recommends the model. In our SSN_GATNE-T model, we use some descriptive symbols. To explain the model more clearly, we use some symbols; Table 1 describes the meaning of each symbol.

In the YouTube dataset, our model collects the connections between nodes and edge types to determine whether there is a connection between users, whether they have shared friends, whether there are the same subscription videos, whether they are all subscribed users of the same video, and connections between them and whether there are five types of interactive information of common favourite videos, and our improved attention mechanism can help us to better label these five interactive relationships. If all five interactive relationships exist in the obtained annotation information, then the users can recommend videos to each other that they have not watched; then there are four users with the same interaction relationship, and then three, two, and the last, *etc.* The most important thing is if two of these five interaction relationships exist between users, their hobbies will be very similar. The potential interaction relationship is obtained through the
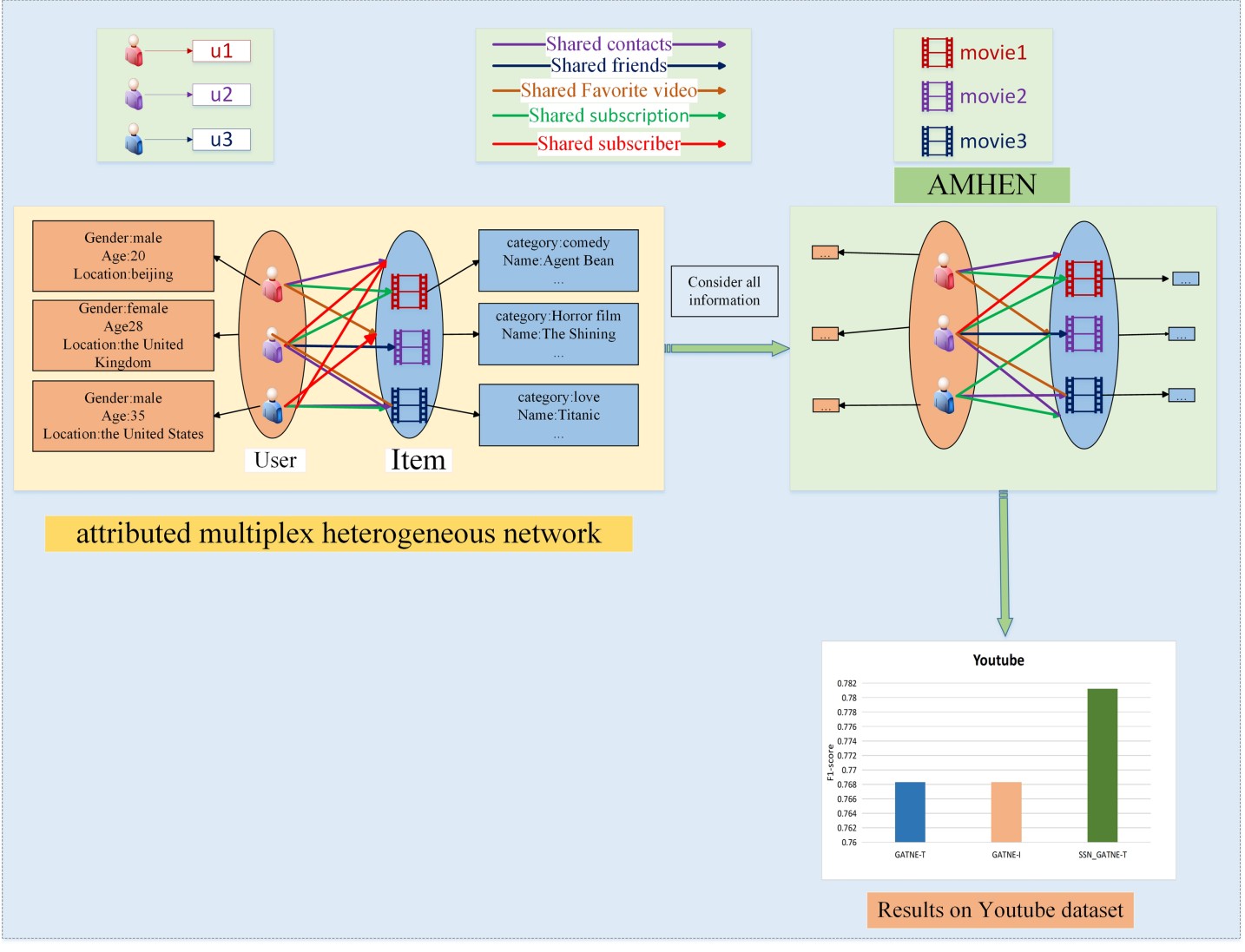

**Figure 1 Recommendation description structure diagram of the YouTube dataset.**

node and edge types, and then the interaction correlation between the two is judged. The correlation between two of the five edge types is inferred, and then five more relevant videos are selected for recommendation. The same is true for the Amazon dataset; the difference is that the variables contained in the dataset are different.

To obtain a more accurate recommendation effect, we can only obtain the best model combination through continuous experiments so that our model can more accurately identify the user-item interaction relationship. Our model can help us better obtain the information of each attribute node in the user-item interaction through an attributed multiplex heterogeneous network, and the improved attention mechanism can help annotation obtain more useful information. To improve the attention mechanism, in this part, we apply the characteristics of the softsign and sigmoid functions to the attention

**Table 1 Notations to describe.**

| Notation | Description |
|---|---|
| G | the input network |
| R | the attribute set of G |
| V | the node set of G |
| $\varepsilon$ | the edge set of G |
| n | the number of nodes |
| r | an edge type |
| v | a node in the graph |
| N | the neighborhood set of a node on an edge type |
| d | the dimension of base/overall embeddings |
| s | the dimension of edge embeddings |
| b, v, c, u | the base/overall/context/edge embedding of a node |

mechanism to obtain more potential hobbies of users that can be accurately recommended for users. The formula is described as follows:

$$a_{i,r} = \text{softsign}\left(w_r^T \text{sigmoid}(W_r U_i)\right)^T \tag{1}$$

$$a_{i,r} = \frac{\left(w_r^T \text{sigmoid}(W_r U_i)\right)^T}{1 + \left|\left(w_r^T \text{sigmoid}(W_r U_i)\right)^T\right|} \tag{2}$$

$$a_{i,r} = \frac{\left(w_r^T \dfrac{W_r U_i}{1 + e^{-(W_r U_i)}}\right)^T}{1 + \left|\left(w_r^T \dfrac{W_r U_i}{1 + e^{-(W_r U_i)}}\right)^T\right|} \tag{3}$$

In addition, in the fully connected layer of the model, we apply the characteristics of the softsign function, which can better reduce the user potential that is discovered through the attributed multiplex heterogeneous network and the improved attention mechanism through the user interaction information for different node attributes. The loss of potential information that can be used for accurate recommendation has achieved good results in obtaining potential information. As a result of the information obtained by using the attribute multiplexing heterogeneous network, the network can more completely obtain the attribute node information in the user-item interaction, and our improved attention mechanism can help us to mark more useful information, mine more potential interests and potential interaction relationships of users, and then provide users with the most matching recommendations based on the potential information that is mined.

Taking the YouTube dataset as an example, we made a structure diagram of the recommendation description process, as shown in Fig. 1. On the left side of Fig. 1, the user and video information is described, and five interaction relationships between the two are outlined. The right part depicts the AMHEN chart setting method. Through this network, the information of each attribute node in the user-item interaction can be better obtained; that is, the potential information existing in two nodes and five edge types can be

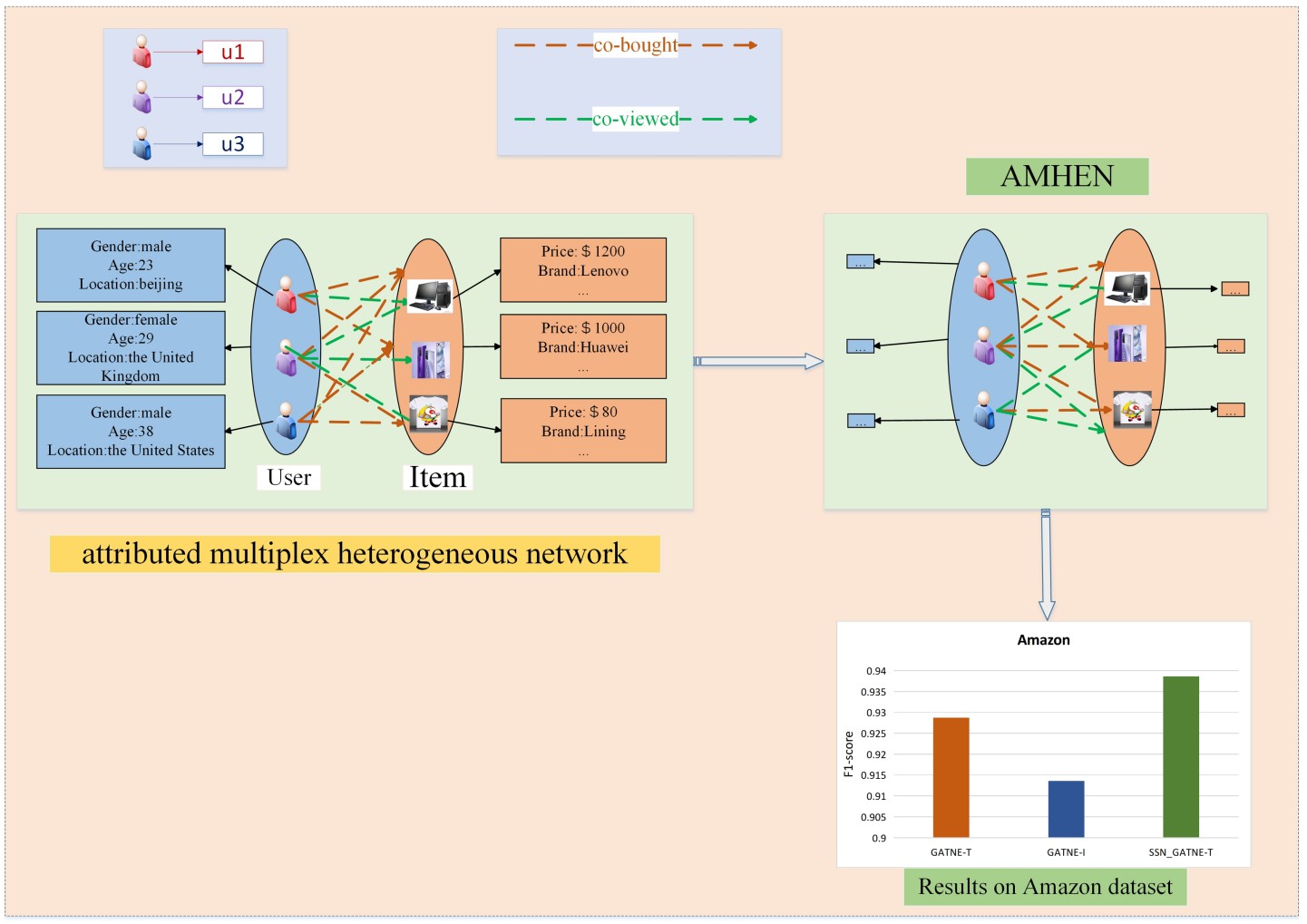

**Figure 2  Recommendation description structure diagram of the Amazon dataset.**

mined, and the improved attention mechanism contained inside can help in tagging obtain more useful information. As more potential information is obtained from users, we have more information to provide users with more accurate recommendations. The lower side of Fig. 1 depicts the recommendation performance comparison of the SSN_GATNE-T model and the GATNE-T and GATNE-I models on the YouTube dataset.

Taking the Amazon dataset as an example, we made a structure diagram of the recommendation description process, as shown in Fig. 2. On the left side of Fig. 2, the user and video information is described, and two interaction relationships between the two are outlined. The right side depicts the setting of the attributed multiplex heterogeneous network (AMHEN) chart. Through this network, the information of each attribute node in the user-project interaction can be better obtained, that is, to explore the potential between the two nodes and the two types of edge types. The two types of edges, such as collaborative purchase and collaborative viewing of information, can help to dig out more

potential interests of users, and the improved attention mechanism used can help labels obtain more useful nodes and edge types.

Sexual information. Since our model obtains and annotates more potential user information, we have more information to provide users with more accurate recommendations. The lower side of Fig. 2 depicts the recommendation performance comparison of the SSN_GATNE-T model and the GATNE-T and GATNE-I models on the YouTube dataset.

Figures 1 and 2 show examples of AMHEN. Figure 1 contains two node types and five edge types. The edge type includes whether there is a connection between users, whether they have shared friends, whether there are the same subscription videos, whether they all subscribe to the same video and whether there are common favourite videos among them. Figure 2 also contains two types of nodes and two types of edge types. The edge types include product attributes and synergy between products as well as view and collaborative purchase links. The two types of nodes in Figs. 1 and 2 are both users and items with different attributes.

## SSN_GATNE-T network structure

The SSN_GATNE-T model is shown in Fig. 3. The application object of this model is an attributed multiplex heterogeneous network. This model uses two parts of "basic embedding" and "edge embedding" to complete the entire embedding, as shown in Fig. 3. The upper and lower parts of the middle are "base embedding" and "edge embedding". The purpose of base embedding is to separate the topological features of the network. Edge embedding is the embedding representation of a specific node on different edge types. Take the orange node as an example. To calculate its edge embedding, there are several types of edges connected to it. From the graph, we can find that there are two types of edges connected to it: green and blue. Then, we separate the green and blue edges for this node. Two new network structures have been created.

In general, the SSN_GATNE-T model does not integrate node attributes for the embedding representation of each node. Instead, the model aggregates neighbouring nodes to generate their own edge embedding representations for different types of edge types connected to the node and then introduces improvements. The attention mechanism calculates their respective attention coefficients (that is, their different importance) to perform a fusion, thereby obtaining the overall embedding representation of the node.

Starting from the embedding learning of attribute-based multiple heterogeneous networks of the transduction environment, our model SSN_GATNE-T is given, as shown in Fig. 3. More specifically, in SSN_GATNE-T, node $v_i$ on edge type r consists of two parts: edge embedding and foundation embedding. Each node $v_i$ type is different and its foundation embedding can be shared. For node $v_i$ on edge type R, its k-level edge embedding $\mathbf{u}_{i,r}^{(k)} \in R^s, (1 \leq k \leq K)$ is aggregated by its neighbour's edge embedding, as follows:

$$\mathbf{u}_{i,r}^{(k)} = \text{aggregator}\left(\left\{u_{j,r}^{(k-1)}, \forall v_j \in N_{i,r}\right\}\right) \tag{4}$$

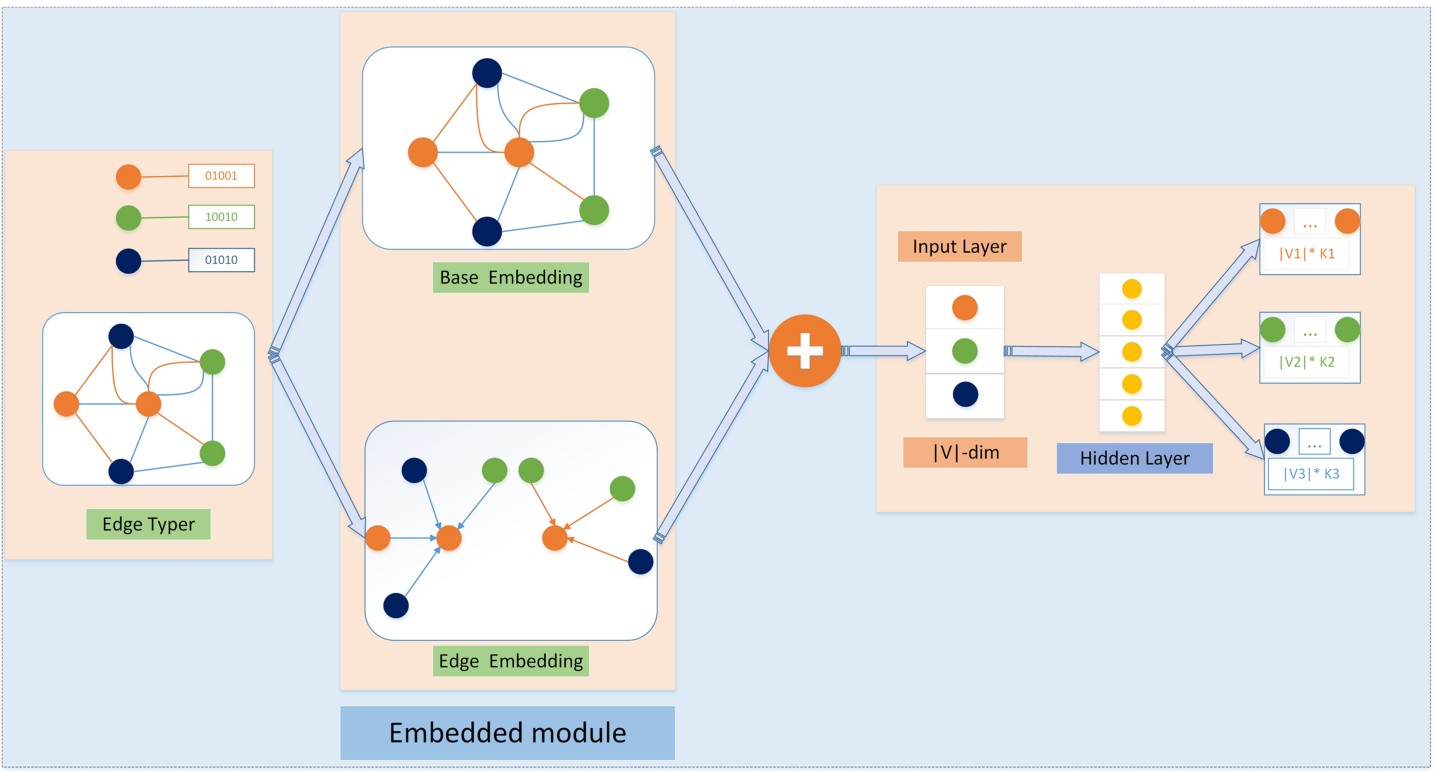

**Figure 3 The SSN GATNE-T model schematic diagram** SSN GATNE-T uses network structure information for embedding with a particular node on each edge type r using base embedding and edge embedding to complete the overall data embedding, while the base embedding between each node isshared across edge types and the edge embedding of a node is achieved by aggregating information about the interactivity around its node, so the output layer of the heterogeneous jump graph specifies a set of polynomial distributions for each node type in the neighbourhood of the input node V. In this example, V = $V_1$ ∪ $V_2$ ∪ $V_3$ and $K_1$, $K_2$, $K_3$ specify the size of the neighbourhood of V on each node type separately.

where $\mathbf{N}_{i,r}$ is the neighbour of node $v_i$ on edge type r. In our direct push model, the initial edge embedding $U_{i,r}^{(0)}$ of nodes and edge types is randomly initialized. Moreover, the mean aggregator can also be used as the aggregation function, as follows:

$$u_{i,r}^{(k)} = \sigma\left(\hat{W}^{(k)} \cdot \text{mean}\left(\left\{u_{j,r}^{(k-1)}, \forall v_j \in N_{i,r}\right\}\right)\right) \tag{5}$$

Or use the max-pool aggregator:

$$\mathbf{u}_{i,r}^{(k)} = \max\left(\left\{\sigma\left(\left\{\hat{W}_{pool}^{(k)} u_{j,r}^{k-1} + \hat{b}_{pool}^{(k)}\right), \forall v_j \in N_{i,r}\right\}\right) \tag{6}$$

Among these functions, edge embedding $v_i$ is represented by the $K^{th}$ level edge embedding $u_{i,r}$, the activation function is represented by, and $U_i$ is formed by embedding and connecting all edges of node $v_i$, and its size is m:

$$U_i = \left(u_{i,1}, u_{i,2}, \ldots, u_{i,m}\right) \tag{7}$$

The self-attention mechanism (*Cen et al., 2019*) is used to calculate the coefficient $\mathbf{a}_{i,r} \in R^m$ of the linear combination of vectors $U_i$ on edge types r, and the softsign function is applied thereto as follows:

$$a_{i,r} = \text{softsign}\left(w_r^T \text{sigmoid}(W_r U_i)\right)^T \tag{8}$$

where $w_r$ and $W_r$ are trainable parameters of edge types of sizes $d_a$ and $d_{a \times s}$, respectively, and the transposition of the matrix is represented by superscript T. Therefore, the overall embedding of node $v_i$ can be expressed as:

$$v_{i,r} = b_i + \alpha_r M_r^T U_i a_{i,r} \tag{9}$$

where $b_i$ is the basic embedding of node $v_i$, $\alpha_r$ is the superparameter representing the importance of edge embedding to overall embedding, and $M_r \in R^{s \times d}$ is the trainable transformation matrix.

When using the attributed multiplex heterogeneous network, we also adopted a self-attention mechanism. The self-attention mechanism we use is a combination of the softsign function and the sigmoid function. That is, $a_{i,r} = \text{softsign}\left(w_r^T \text{sigmoid}(W_r U_i)\right)^T$, The combination of the improved attention mechanism and the attribute multiplexing heterogeneous network has a very good effect on the three evaluation indexes of ROC-AUC, PR-AUC, and F1, while the improvement of the F1 index is greater. In addition, we also chose the softsign function of the activation function of the fully connected layer of the model because we found that using the softsign function here combined with the attention mechanism we adopted has unexpected results, which greatly improves the evaluation index F1, showing the effectiveness of the recommendation performance of our model. In order to better to describe our SSN_GATNE-T model, we describe our SSN_GATNE-T algorithm in Algorithm 1.

## Model architecture

The SSN_GATNE-T model combines attributed multiplex heterogeneous networks with an attention mechanism that introduces softsign and sigmoid function properties attributed multiplex heterogeneous network can help to obtain more attributes of the user-item information, regardless of the number of nodes and type of model, our model can be good at handling, and improved attention mechanism can help labeling to obtain more useful information, the combination of the two can help to explore more potential information to improve the recommendation effect. In addition, the application of the softsign function in the fully connected layer of the model can better reduce the loss of potential information about the user that can be used for accurate recommendations, and we use the Adam optimizer to optimize our model in order to avoid the impact on the accuracy of the model recommendations due to the excessive node and edge type information obtained. The optimisation of the model can qualitatively improve the recommendation effect of the recommender system. The optimization methods include Adagrad, Momentum, RMSprop, GradientDescent and Adam. Through extensive experimental validation, we use the Adam optimiser for the model optimisation,

---

> **Algorithm 1  SSN_GATNE-T.**
>
> **Inputs:** Network $G = (V, \xi, A)$, base embedding dimension d, edge embedding dimension s, learning rate, coefficients $\alpha, \beta$.
>
> **Output:** the overall embedding $v_{i,r}$ of all different nodes v with different edge types r
>
> **1** Initialise all model parameters
>
> **2** Generate a random node sample $(v_i, v_j, r)$ of edge type r
>
> **3** Generate training sample $\{(v_i, v_j, r)\}$ by random samples related to edge type r.
>
> **4** When not converged
>
> **5**      For each $(v_i, v_j, r) \in$ training sample
>
> **6**      Use equation (9) to calculate the overall embedding $v_{i,r}$
>
> **7**      Update the model parameters

which allows for faster convergence and also helps with the tuning of the model. The Adam optimizer is shown in the following equation.

$$m^t = \mu * m_{t-1} + (1 - \mu) * g_t \tag{10}$$

$$n_t = v * n_{t-1} + (1 - v)^* g_t^2 \tag{11}$$

$$\hat{m}_t = \frac{m_t}{1 - v^t} \tag{12}$$

$$\hat{n} = \frac{n_t}{1 - v_t} \tag{13}$$

$$\Delta\theta_t = -\frac{\hat{m}_t}{\sqrt{\hat{n}_t} + \varepsilon} * \eta \tag{14}$$

# EXPERIMENT

In this part, we first introduce the setting of two evaluation data sets and parameters; second, we compare our recommendation algorithm with other state-of-the-art methods to verify the effectiveness of our recommendation model and conduct a bar graph and a line graph. By comparison, we can clearly see the improvement of the three evaluation indexes of ROC-AUC, PR-AUC, and F1 by our proposed method. Finally, we analyse each step of the improvement of our model through ablation experiments to verify the effectiveness of our model.

## Dataset and parameter settings

In this article, we use YouTube and Amazon datasets to complete the experiment. The Amazon product data set includes product metadata and links between products; the YouTube data set includes various types of interactions. Since some baselines cannot be extended to the entire graph and the total data volume of nodes and edges contained in the original data set is too large, we selected part of the data in the original data set as the sampling data, which is the sampling data set we adopt, and pass SSN_GATNE-T. The model evaluates the recommendation effect and performance of the sampled data set. Table 2 describes the information of the two sampled datasets, including the total number

**Table 2 Dataset statistics.**

| Dataset | Amazon | Youtube |
|---------|--------|---------|
| nodes | 10,116 | 2,000 |
| edges | 148,865 | 1,310,617 |
| n-types | 1 | 1 |
| e-types | 2 | 5 |

of nodes, the total number of edges, the number of node types, and the number of edge types. The n-type and e-type in Table 2 represent the node type and edge type, respectively.

**Amazon.** In the experiment, the two interactions of the product attributes contained in the data and the collaborative viewing between the products and the collaborative buying link were used, which can be observed and understood from Fig. 2.

**YouTube.** The YouTube dataset contains five types of interactions, including shared friends and shared subscriptions, which can be observed from Fig. 1.

In the model, to filter out the most suitable parameter settings for the SSN_GATNE-T model, we chose to experiment the settings and determine the parameters one-by-one. Taking the embedding size of the whole and the edge as an example, the number of parameters with setting changes are only two. The settings of other parameters do not change during the process of determining these two parameters, and the determination of these two parameters is to first determine one of them and then determine the other while maintaining the determined items. After many tuning experiments, we finally set the embedding size of the whole and the edge to 200 and 10, respectively; the walking length and times of the node to 10 and 20, respectively; and the window to 5. The number of negative samples L of each positive training sample fluctuates in the range of 5-50. For each edge type r, coefficients $\alpha_r$ and $\beta_r$ are set to 1. In the experiment, if the ROC-AUC of the verification set reaches the optimal value before all iterations are completed, the model will be terminated early.

## Evaluation index

In this article, to better understand the recommended performance of the model, we use the ROC-AUC, PR-AUC and F1 (F1-score), which are three evaluation indicators.

The ROC-AUC is an evaluation index that comprehensively considers the ROC and AUC. The two indexes of sensitivity and specificity are not affected by unbalanced data.

ROC-AUC makes an evaluation indicator that takes into account both ROC and AUC. F1 is an evaluation indicator that takes into account both precision and recall values.

F1 is an evaluation index that fully considers the precision value and the recall value:

$$F-\text{score} = \frac{2}{\frac{1}{P} + \frac{1}{R}} = \frac{2PR}{P + R} \qquad (15)$$

where P stands for Precision and R stands for Recall.

$$predision = \frac{tp}{tp + fp} \tag{16}$$

$$recall = \frac{tp}{tp + fn} \tag{17}$$

To solve the single-point limitation of the P, R, and F-measures, we use PR-AUC/AP (average precision), which can reflect the overall evaluation index. The larger the PR-AUC value is, the better the performance of the model.

$$PR - AUC/AP = \int_0^1 p(r)d(r) \tag{18}$$

## Comparison of model accuracy

In this section, we compare our model SSN_GATNE-T with ANRL, PMNE (n) (*He et al., 2021*), PMNE(r), PMNE(c), MVE(Multi-View Network) (*Berahmand, Nasiri & Li, 2021*), MNE (*Li et al., 2021*), GATNE-T, GATNE-I, FAME, FANEm and other models for the evaluation of the Youtube and Amazon datasets, and express them in line graph to analyse in detail the impact on our model on these two datasets and where its advantages lie. For this model comparison, we demonstrate the effectiveness of our model for large network data by comparing the SSN_GATNE-T model with other models on two large network datasets containing different nodes and different edge types, Amazon and Youtube, and then demonstrate that the potential information on the different edge types of user nodes and project nodes obtained by improving the performance of the model for large networks can be used to improve the effectiveness of the recommendation performance.

In the experiment, we divided the dataset into a training set, a validation set and a test set and used the area under the ROC curve (ROC-AUC), the area under the PR curve (PR-AUC) and the F1 score as the evaluation indicators for model evaluation. The accuracy of our model and the pros and cons of its recommendation effect are judged through the impact of these three evaluation indexes on the two YouTube and Amazon datasets *via* different models, as shown in Table 3. When evaluating, we compare our model SSN_GATNE-T with models including metapath2vec, ANRL, PMNE(n), PMNE(r), PMNE(c), MVE, MNE, GATNE-TGATNE-I, FAME and FANEm. Our model has different effects on different datasets, and each model has different effects on these three evaluation indexes. Compared with other models, our model SSN_GATNE-T aggregates neighbour nodes to generate their own edge embedding representations for different types of edge types connected to nodes and then introduces an improved attention mechanism to calculate their respective attention. The coefficient to performs a fusion, thereby obtaining the overall embedding representation of the node. Compared with other models, our model can better obtain the interaction relationship between nodes and edge types, thereby digging out more potential user interests. In addition, our improved attention mechanism can also help us obtain and label node and edge types.

**Table 3 Comparison of evaluation results of various models.**

| | YouTube | | | Amazon | | |
|---|---|---|---|---|---|---|
| | ROC-AUC | PR-AUC | F1 | ROC-AUC | PR-AUC | F1 |
| DeepWalk | 0.7111 | 0.7004 | 0.6552 | 0.942 | 0.9403 | 0.8738 |
| node2vec | 0.7121 | 0.7032 | 0.6536 | 0.9447 | 0.943 | 0.8788 |
| LINE | 0.6424 | 0.6325 | 0.6235 | 0.8145 | 0.7497 | 0.7635 |
| metapath2vec | 0.7098 | 0.7002 | 0.6534 | 0.9415 | 0.9401 | 0.8748 |
| ANRL | 0.7593 | 0.7321 | 0.7065 | 0.7168 | 0.703 | 0.6772 |
| PMNE(n) | 0.6506 | 0.6359 | 0.6085 | 0.9559 | 0.9548 | 0.8937 |
| PMNE(r) | 0.7061 | 0.6982 | 0.6539 | 0.8838 | 0.8856 | 0.7967 |
| PMNE(c) | 0.7039 | 0.6982 | 0.6539 | 0.9355 | 0.9346 | 0.8642 |
| MVE | 0.7039 | 0.7010 | 0.6510 | 0.9298 | 0.9305 | 0.8780 |
| MNE | 0.8230 | 0.8218 | 0.7503 | 0.9028 | 0.9174 | 0.8325 |
| GATNE-T | 0.8461 | 0.8193 | 0.7683 | 0.9744 | 0.9705 | 0.9287 |
| GATNE-I | 0.8447 | 0.8232 | 0.7683 | 0.9625 | 0.9477 | 0.9136 |
| FAMEm | – | – | – | 0.939 | 0.937 | 0.887 |
| FAME | – | – | – | 0.959 | 0.950 | 0.900 |
| SSN_GATNE-T | 0.8583 | 0.8437 | 0.7812 | 0.9754 | 0.9717 | 0.9386 |

The potential correlation information between the node and edge types is precisely because our model can obtain more important information that can be used for recommendations compared with other models, which makes our model have a great improvement in recommendation performance. Compared with the DeepWalk, LINE, node2vec, NetMF, and GraphSAGE network embedding models, our SSN_GATNE-T model can obtain not only node information but also the potential interaction relationship between nodes and edge types and the relationship between nodes and edges. The more types and quantities there are, the more options that can be used to provide users with accurate recommendations. Compared with HEN, PTE, metapath2vec and HERec, which are heterogeneous network embedding methods, our model mines the potential information in the network more easily than the heterogeneous network, which is full of people, to obtain the combination of content and structure. It also makes a great contribution to the scalability of the network. Our model can not only be used for the processing of small networks but can also perfectly mine its potential information for large networks containing billions of nodes and edge types. Compared with PMNE, MVE, MNE, and Mvn2vec, which are eager for multichannel heterogeneous network embedding methods, compared to the limitation of capturing a single view of the network and the effectiveness of obtaining node and edge type embedding information, our model is superior; the edge type correlation information mining and labelling are more comprehensive. Compared with TADW, LANE, and ANRL, which are these types of attribute network embedding methods, attribute network embedding is a low-dimensional vector representation of nodes looking for and retaining the network topology and the proximity of node attributes, and our model can not only do this, it is also possible to mine

more potential attribute information for recommendation through the correlation between node and edge types. Compared with the existing recommendation algorithms, our SSN_GATNE-T model can either obtain the correlation information between labelled nodes and edge types, obtain the attribute information of nodes and edge types, or process billions of nodes and edges. In the processing of large-scale networks of nodes and edges, our model can use more or more relevant information for recommendations, and the recommendation effect is greatly improved.

Our SSN_GATNE-T model has the most obvious improvement effect on the YouTube dataset, and it has a corresponding improvement for Amazon. Although both of these datasets contain two nodes, users and projects, because the Amazon dataset contains only two edge-type interactions, while the YouTube data set contains five edge-type interactions, the nodes contained in the two data sets are related to the number of edges being different, which is also the reason for the different recommendation effects of the two. Since there are five types of edges in the YouTube dataset and there are only two types of edges in the Amazon dataset, our SSN_GATNE-T model can mine more user potential information for recommendations in the correlation between nodes and edge types. Yes, compared to other advanced recommendation models, our model shows a greater improvement on the YouTube dataset than on the Amazon dataset. However, the YouTube dataset contains a total of 2,000 nodes and 1,310,617 edges, and the Amazon dataset contains 10,116 nodes and 148,865 edges. In contrast, the Amazon dataset contains more nodes but fewer edges. The network constructed by the Amazon dataset is simpler than the network constructed by the YouTube dataset in the information it obtains and annotates. Compared with the YouTube dataset, there is less irrelevant information to be excluded. Although the improvement in the Amazon dataset is less than the improvement in the YouTube dataset compared to other models, the final recommendation effect is the best. Yes, we can conclude from its evaluation indexes ROC-AUC, PR-AUC, and F 1 that our model reached 0.9754, 0.9717, and 0.9386, respectively, while only reaching 0.8583, 0.8437, and 0.7812 on the YouTube dataset. In addition, our model has improved the three evaluation indexes of ROC-AUC, PR-AUC, and F1, but the improvement for F1 is more obvious. To this end, we have made separate line graphs of the evaluation metrics of our model and other models for the Amazon and Youtube datasets, as shown in Figs. 4 and 5, to give you a clearer picture.

For the Youtube dataset, it contains user nodes and project nodes, with the user nodes having attributes such as gender, age and address, and the project inputs having information such as movie name and movie genre, and also consisting of five edge types of interaction between different nodes. The user's hobbies and interests obtained from different interactions in the YouTube dataset are also different and should be treated differently. Our model SSN_GATNE-T will not ignore these node attribute user-item interactions. Compared with the four models (DEEPWalk, metapath2vec, MVE, and PMNE(c)), our model pays more attention to the attributes of nodes and user-item interactions. The five interaction relationships that exist in the YouTube dataset will affect our model. The recognition rate of user-item interactions has increased. Heterogeneous networks can collect more information between users through attribute reuse, and a
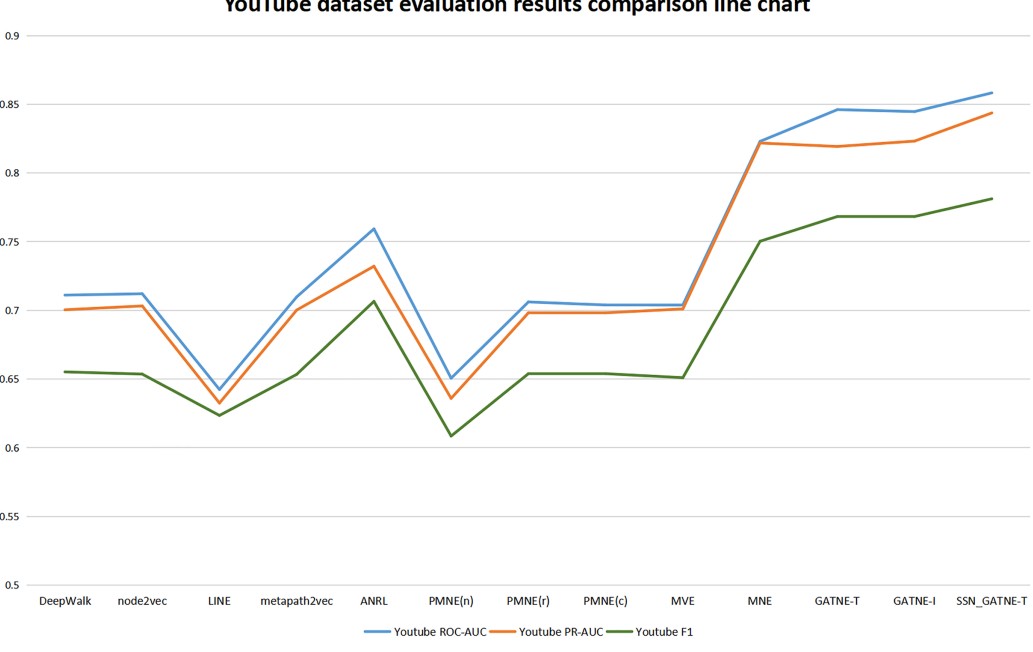

**Figure 4 YouTube dataset evaluation results comparison line chart.**

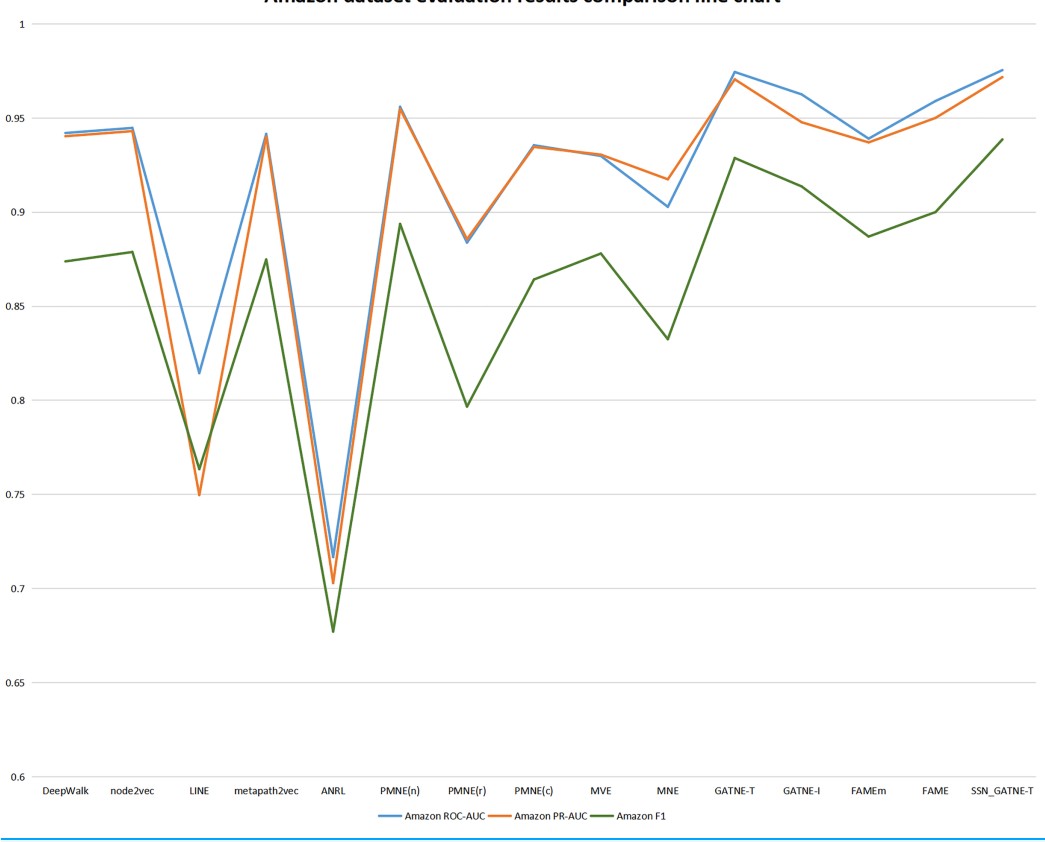

**Figure 5 Amazon dataset evaluation results comparison line chart.**

self-attention mechanism can better mark and obtain the information we need to make more changes for users as good and more suitable recommendations, so its evaluation index has been greatly improved. In comparison with the observations in Table 3, the evaluation index of the LINE model is not very good. Compared with the LINE model, ROC-AUC has improved by +33.61%, PR-AUC by +33.39% and F1 by +25.29%. All three assessment indices of the SSN_GATNE-T model gained a greater or lesser improvement compared to other models outside the LINE model. Therefore, our model SSN_GATNE-T has a good recommendation effect on the YouTube dataset and has a good effect on the three evaluation indexes.

The Amazon dataset includes collaborative viewing and collaborative purchasing links between product attributes and products, which are often overlooked. Our model SSN_GATNE-T does not ignore these node attribute user-item interactions. Compared with the four models DEEPWalk, metapath2vec, MVE, and PMNE(c), our model pays more attention to the node attributes and user-item interactions. Since the interaction relationship contained in the Amazon data set is only the collaborative viewing and collaborative purchase links between product attributes and products, the four models of DEEPWalk, metapath2vec, MVE, and PMNE(c) have good evaluation results on the Amazon data set. Therefore, its evaluation index is less effective than YouTube, a dataset with five interaction relationships. However, our model is still greatly improved compared to the four models of DEEPWalk, metapath2vec, MVE, and PMNE(c). In comparison with the observations in Table 3, the evaluation index of the ANRL model is not very good. Its recommended performance is the worst. Compared with the ANRL model, the SSN_GATNE-T model has increased ROC-AUC by +36.08%, PR-AUC by +38.21%, and F1 by +38.60%. All three assessment indices of the SSN_GATNE-T model received a greater or lesser improvement compared to other models outside the ANRL model. Therefore, our model SSN_GATNE-T has a good recommendation effect on the Amazon dataset, especially the evaluation index F1.

For large network datasets using network embedding methods of the acquisition of correlation information about different nodes and different edge types the models that work relatively well are GATNE-T, GATNE-I, FAME, FANEm and our SSN_GATNE-T model, the evaluation indices of each model are shown in Table 3. The GATNE-T and GATNE-I models are accomplished by dividing the overall embedding of the model into base embedding and edge embedding, and the GATNE-I model also incorporates attribute embedding. Compare with the GATNE-T and GANTE-I models, the edge embedding of SSN_GATNE-T is more powerful because our improved attention mechanism can better aggregate the neighbourhood node information and then obtain more relevant information between the node and edge types for recommendation to improve the accuracy of the recommendation model. Compared to the GATNE-T and GATNE-I models, the SSN_GATNE-T model obtained significant improvements in all three evaluation indices, ROC-AUC, PR-AUC, and F 1, especially the F1 index, for the Amazon and YouTube datasets. On the Amazon dataset, the assessment index F1 improved by +1.07% compared to GATNE-T and by +2.74% compared to GATNE-I; on the YouTube dataset, the assessment index F1 improved by +1.68% compared to

GATNE-T and by +1.68% compared to GATNE-I. The FAME and FAMEm models to achieve fast and effective node representation learning by cleverly integrating spectral graph transformations and node attribute features into a sparse random projection system, and although it takes less time to obtain interaction information on nodes and edge types than the SSN_GATNE-T model, the number of interactions it obtains is less than that of the SSN_GATNE-T model. The ultimate goal of acquiring information is to improve the effectiveness of the recommendation performance, and the SSN_GATNE-T model is more capable of acquiring information about potential interactions between different nodes and edge types, which in turn leads to more accurate recommendations. Compared to the FAME and FAMEm models on the Amazon dataset, the SSN_GATNE-T model also achieved significant improvements in the ROC-AUC, PR-AUC and F 1 evaluation indices, with the ROC-AUC improving by 1.71% and 3.88% respectively, the PR-AUC improving by 2.28% and 3.70% respectively, and the F 1 improving by 4.29% and 5.82% respectively. The ROC-AUC increased by 1.71% and 3.88%, the PR-AUC by 2.28% and 3.70% and the F 1 by 4.29% and 5.82%. This can be verified from the comparative information on the evaluation indices between the models in Table 3. Therefore, compared to the GATNE-T, GATNE-I, FAME and FAMEm models, our SSN_GATNE-T model, both for the Amazon dataset and the Youtube dataset, is able to aggregate the interaction information between different edge types of neighbouring nodes of each node very well, and due to the stronger edge embedding capability of our model, we can obtain more comprehensive. The potential interaction information between different nodes and edge types can be more comprehensively captured and used only for recommendation, thus improving the effectiveness of the model's recommendation performance.

In order to get a clearer picture of the strengths and weaknesses for each model, we have made line graphs of the evaluation results of each model for the Amazon and Youtube datasets. These are shown in Figs. 4 and 5.

In order to get a clearer picture of the strengths and weaknesses for each model, we have made line graphs of the evaluation results of each model for the Amazon and Youtube datasets. These are shown in Figs. 4 and 5. The rising trend of the line graph shows the effectiveness of the SSN_GATNE-T model in processing the data onto the large network model in the Amazon dataset, which can further show that the SSN_GATNE-T is more efficient in acquiring information about the interactions between different nodes and different edge types, and the information obtained is more comprehensive, which in turn improves the effectiveness of the recommendation performance when the information obtained is used for recommendation. The evaluation index of our model SSN_GATNE-T is at the highest point of the line graph, which shows that our model has indeed improved. This shows that our model is more effective than other mainstream recommendation models in processing and analysing the YouTube and Amazon datasets and in making recommendations, which is a testament to the effectiveness of our model.

Since the Adam optimisation approach allows the model to converge faster and then achieve accurate recommendations faster and better, and is also very helpful for model tuning, we have adopted the Adam optimization approach from/to the SSN_GATNE-T

**Table 4 Comparison of model optimization approaches.**

| | Youtube | | | Amazon | | |
|---|---|---|---|---|---|---|
| | ROC-AUC | PR-AUC | F1 | ROC-AUC | PR-AUC | F1 |
| Adagrad | 0.5549 | 0.5546 | 0.5452 | 0.5849 | 0.5510 | 0.5632 |
| Momentum | 0.8256 | 0.8327 | 0.7677 | 0.9547 | 0.953 | 0.9205 |
| RMSprop | 0.8435 | 0.8365 | 0.7796 | 0.9685 | 0.9657 | 0.9302 |
| GradientDescent | 0.8297 | 0.7977 | 0.7500 | 0.9726 | 0.9675 | 0.9293 |
| Adam | 0.8583 | 0.8437 | 0.7812 | 0.9754 | 0.9717 | 0.9386 |

model. In optimizing the model, we used not only Adam's optimisation method but also Adagrad, Momentum, RMSprop and GradientDescent to optimize the model, and the evaluation indices of the recommendation system after optimization is shown in Table 4. Although each optimizer has its own advantages and disadvantages for different datasets, there is still a gap compared to the Adam optimization method, and the network and each parameter of the model are the same when choosing the model optimization method, so the Adam optimization method is chosen for our SSN_GATNE-T model.

## Fridman test

In order to better describe the effectiveness of the model, we use the Fridman algorithm to analyse the effectiveness of our SSN_GANTE-T model in obtaining relevance information of small and large networks containing multiple types of nodes and multiple types of edges through attributed multiplex heterogeneous network and an improved attention mechanism, and verify the effectiveness of our model in obtaining the missing information on the interaction between different nodes and different edge types in a large network of hundreds of millions of nodes and the effectiveness of obtaining potential information, and then the effectiveness of the obtained information in improving the accuracy of recommendations when used in a recommendation system.

For both the Youtube and Amazon datasets, we defined four variables, edge_type, node1, node2 and label, to perform non-parametric tests on the results we obtained to check the relevance of the interest implied. node1 and node2 represent user nodes and item nodes respectively. Whether it is the YouTube dataset or the Amazon dataset, we have defined four variables, edge-type, node1, node2, and label, to perform nonparametric tests on the results we obtained to test the relevance. During the analysis process, the five types of interactions of edge-type in the YouTube dataset, including shared friends and shared subscriptions, are represented by 1, 2, 3, 4, and 5, respectively. For the edge-type product attributes in the Amazon dataset and the collaboration between products, the two interactions of view and collaborative purchase links are represented by 1 and 2, respectively, and the label is represented by 1 and 0 to indicate yes or no. For the correlation information between the nodes existing in these two datasets and different types, we use edge-type and label to perform nonparametric tests on the relevant samples of node1 and node2. To this end, we performed a hypothesis test summary and related samples through the Wilcoxon signed rank test summary. Tables 5 and 6 describe the

**Table 5  YouTube dataset assumptions.**

**Hypothesis test summary**

| | Original hypothesis | Test | Significance[a,b] | Decision |
|---|---|---|---|---|
| 1 | The median of the difference between node1 and node2 is equal to 0. | Wilcoxon Signed Rank Test for Correlated Samples. | 1.000 | Keep the original hypothesis. |

**Notes:**
[a] The significance level is 0.050.
[b] Shows progressive significance.

**Table 6  Wilcoxon signed rank test for relevant samples of YouTube dataset.**

| | |
|---|---|
| TotalN | 262,010 |
| Test statistics | 25,120,694,864.000 |
| Standard error | 0.000 |
| Standardized test statistics | 0.000 |
| Progressive significance (two-sided test) | 1.000 |

**Table 7  Hypothesis testing of Amazon data set.**

**Hypothesis test summary**

| | Original hypothesis | Test | Significance[a,b] | Decision |
|---|---|---|---|---|
| 1 | The median of the difference between node1 and node2 is not equal to 0. | Wilcoxon Signed Rank Test for Correlated Samples. | 1.000 | Reject the original hypothesis. |

**Notes:**
[a] The significance level is 0.050.
[b] Shows progressive significance.

**Table 8  Wilcoxon signed rank test for relevant samples of Amazon dataset.**

| | |
|---|---|
| TotalN | 29,488 |
| Test statistics | 234,912,151.000 |
| Standard error | 1,461,801.487 |
| Standardized test statistics | 11.985 |
| Progressive significance (two-sided test) | 0.000 |

test of the YouTube dataset, and Tables 7 and 8 describe two items of the Amazon dataset. In Tables 5 and 6, which describe the summary statistics of the model on the Youtube dataset, we performed the relevant sample Wilcoxon signed rank test of the model results in the median of the difference between node1 and node2 equal to 0 as the original hypothesis. Test. The summary statistics of the model on the Youtube dataset are described in Tables 4 and 5. We used the median of the difference between node1 and node2 equal to 0 as the original hypothesis to perform the relevant sample Wilcoxon

signed rank test on the model results. The total number of samples tested was 262,010, the test statistic was 25,120,694,864.000 and the significance level a was 0.05, when $P > 0.05$, the original hypothesis is retained, otherwise the original hypothesis is rejected. During the testing of the Youtube dataset, the significance P obtained from the test was 1.000 and the asymptotic significance P from the two-sided test was also 1.000, so the original hypothesis was retained. While Tables 7 and 8 depict the summary statistics of the model on the Amazon dataset, we performed the relevant sample Wilcoxon signed rank test on the model results with the original hypothesis that the median of the difference between node1 and node2 is not equal to 0. The total number of samples tested was 29,488, the test statistic was 234,912,151.00, the significance level a was 0.05, and the significance P obtained from the test is 0. The asymptotic significance P after the two-sided test is also 0, so the original hypothesis is rejected. In the hypothesis testing of the Youtube and Amazon datasets, we also used the Bonferroni adjustment with a more stringent significance level of 0.025 and 0.005 in order to reduce the error in the test results, and the results were the same as those obtained with a significance level of 0.05. The results are the same as those obtained by using a significance level of 0.05. It can be seen that our model provides a stronger boost to the recommendations provided by the processing analysis of the Youtube dataset compared to the Amazon dataset.

Taking the YouTube dataset as an example, we use edge-type and label as the basis to describe the correlation between node1 and node2 through the five interactions they contain and create a continuous field of node1 and node2 that changes with the frequency and total number of nodes. The information histogram is shown in Figs. 6 and 7.

Taking the Amazon dataset as an example, we processed the model on a case-by-case basis, as shown in Table 9. During this process, the missing values of the user and the system are both 0. In our model, this is due to the introduction of attribute reuse heterogeneous network, which indicates the problem of missing information in the process of obtaining potential correlation information between nodes and edge types, making the interactive information we obtain more comprehensive and showing the effectiveness of our model. In addition, we describe the distribution parameters for the estimation of the model as shown in Table 10. We also described the normal P-P graph and detrended normal P-P graph of node2 through edge_types based on node1 and label, as shown in Figs. 8 and 9.

The data and graphs obtained through the Friedman test show that our model has a very obvious effect on the acquisition of the correlation information between each node and different edge-types in the large and small networks, and more potential related information is obtained; moreover, the more effectively we can make accurate recommendations for users. Taking YouTube as an example, our model can identify and obtain more of the five interaction relationships between users and projects: sharing friends, sharing subscriptions, sharing subscribers, and sharing favourite videos, showing the effectiveness of the recommendation effect of our model.

The Friedman test, on the one hand, shows that our SSN_GATNE-T models/modelled does not suffer from missing node information due to the variety and number of nodes and edge types it contains when dealing with large web datasets such as Amazon and

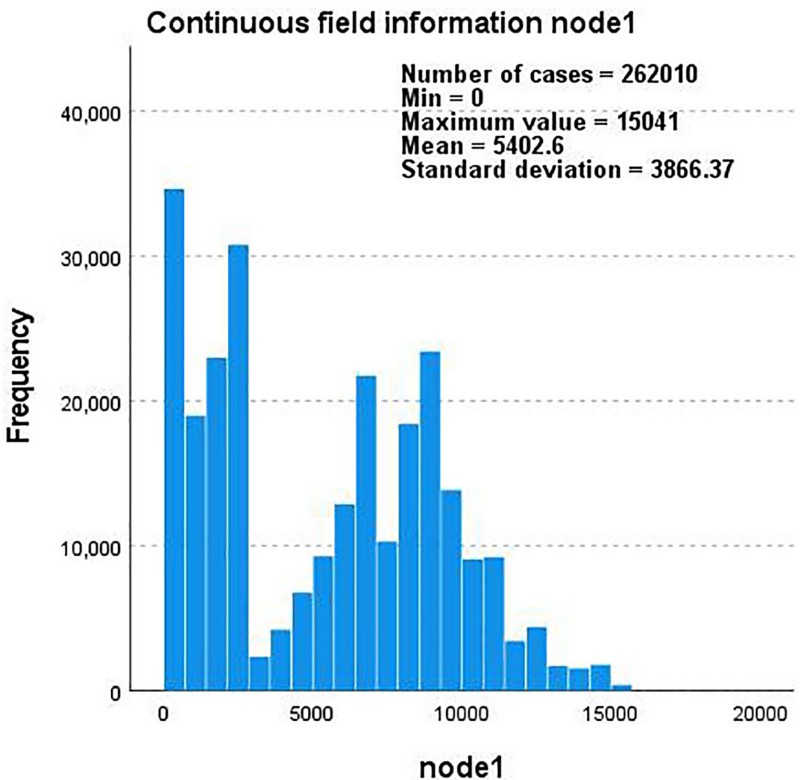

**Figure 6  A histogram of continuous field information for node1 as the frequency and total number of nodes change.**               

Youtube, as evidenced by the zero missing user and item values. On the other hand, the summary of the Wilcoxon signed-rank test and the summary of the node1 and node2 case processing, as well as the estimation of the distribution parameters, demonstrate that the model is more comprehensive in obtaining information about the potential interactions between different nodes and different edge types. Both the absence of missing nodes in the user's project and the availability of more comprehensive information about the potential interactions between different nodes and different edge types can improve the effectiveness of the recommendation performance by providing the user with more relevant information for the recommendation.

## Ablation experiment

In this research, we conducted many experiments on the YouTube and Amazon datasets and obtained the best model settings by improving the model step by step. In the experiments of this article, to obtain more effective information about users and projects and identify more potential interests of users, we first set the parameters of the entire model. The overall and edge-embedding sizes are set to 200 and 10, respectively, the node walking length and number of times are set to 10 and 20, respectively, and the window is set to 5. The number of negative samples L of each positive training sample fluctuates in the range of 5–50. For each edge type r, the coefficients $\alpha_r$ and $\beta_r$ are set to 1. In addition, the model in this paper is optimized by the Adam optimizer. To prove the

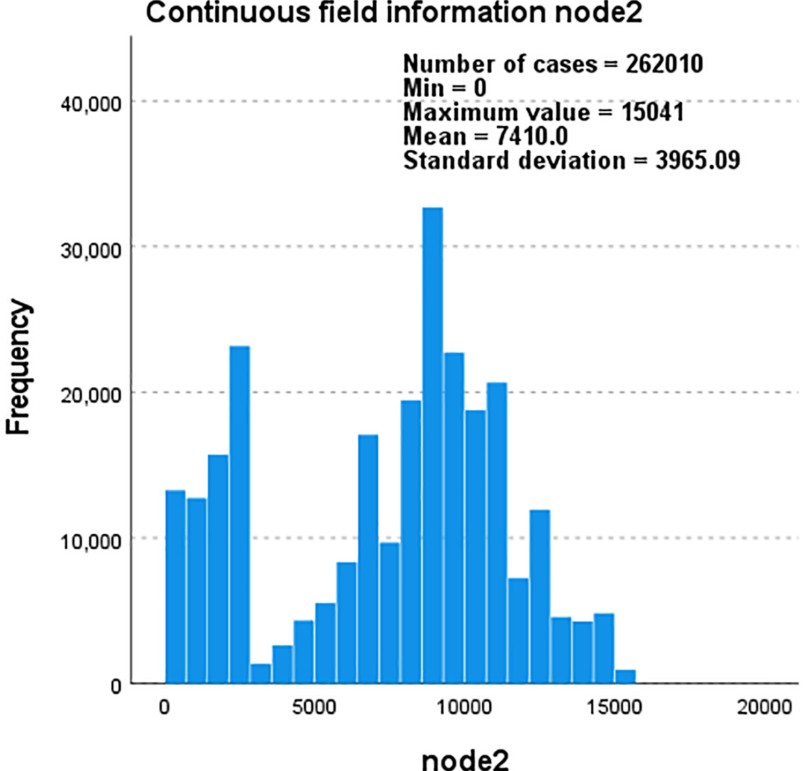

**Figure 7 Histogram of continuous field information for node1 as the frequency and total number ofnodes change.**

**Table 9 Case handling.**

**Case handling summary**

|  |  | node2 | label |
|---|---|---|---|
| Series or sequence length |  | 29,492 | 29,492 |
| Number of missing values in the graph | User missing values | 0 | 0 |
| The cases are not weighted. | System missing value | 0 | 0 |

**Table 10 Estimation of distribution parameters.**

|  |  | node2 | Label |
|---|---|---|---|
| Normal distribution | Position | 332,795.29 | 0.50 |
| The cases are not weighted. | Scaling | 145,311.429 | 0.500 |

validity of the model proposed in this study, we performed ablation analysis on the model itself and compared this model with the model already proposed.

For the hyper parameters of the SSN_GATNE-T model we chose epoch and batch size to tune the accuracy of the model recommendations. When the batch size and other parameters are fixed, the model is tuned by adjusting the epoch to 10, 30, 50, 100, 150 and

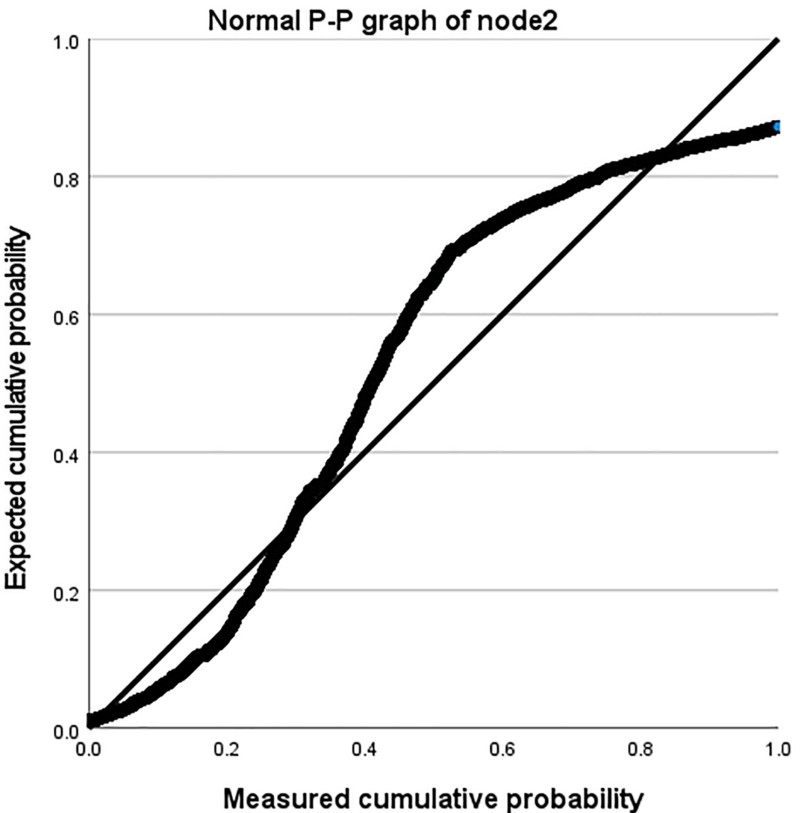

**Figure 8** **Normal P-P diagram of node2.**

200, respectively, to run the model. It was found that the model was not optimal at the end of the run for epoches of 10, 30 and 50, whereas for epoches of 150 and 200, the model reached the optimal solution earlier and then stopped, but the model evaluation results was smaller than the optimal solution obtained for epoch 100. Therefore, if the epoch is chosen to be 100, the final result of the model will be obtained to the optimal solution, neither because the number of iterations is insufficient to reach the optimal solution nor because the model reaches the optimal solution early, which causes the model to be terminated early and the result is not accurate. When epoch was 100 and other parameters were fixed, we determined the size of batch-size to be 64 through experiments, and we also chose batch size of 16, 32 and 128 to adjust the model. The results obtained are not as good as those obtained when the batch size of 128 is chosen, although the convergence time is faster and the training times are faster than that of 64, the accuracy is lower and the evaluation results obtained are not satisfactory. Therefore, the hyper parameters epoch and batch size of SSN_GATNE-T model are 100 and 64 respectively.

In the ablation experiments, we performed the ablation study by deleting or replacing the modules in this model when the epoch and batch size was 100 and 64 respectively and the corresponding parameters were kept constant. In these two data sets, only the attributed multiplex heterogeneous network and the original attention mechanism (GATNE-T) are introduced, and the attributed multiplex heterogeneous network and our

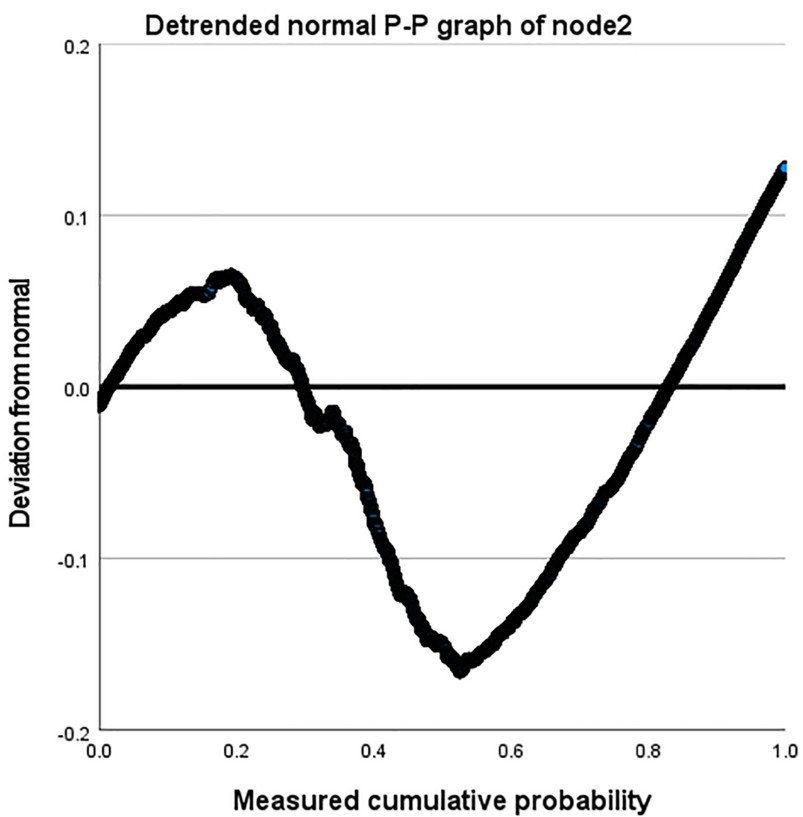

**Figure 9  Detrended normal P-P diagram of node2.**  

**Table 11  Ablation studies on YouTube and Amazon datasets.**

|  | Youtube | | | Amazon | | |
|---|---|---|---|---|---|---|
|  | ROC-AUC | PR-AUC | F1 | ROC-AUC | PR-AUC | F1 |
| GATNE-T | 0.8461 | 0.8193 | 0.7683 | 0.9744 | 0.9705 | 0.9287 |
| S_GATNE-T | 0.8501 | 0.8273 | 0.7702 | 0.9692 | 0.9711 | 0.9281 |
| SN_GATNE-T | 0.8536 | 0.8356 | 0.7776 | 0.9747 | 0.9698 | 0.9343 |
| SSN_GATNE-T | 0.8583 | 0.8437 | 0.7812 | 0.9754 | 0.9717 | 0.9386 |

improved attention mechanism that introduces the softsign and sigmoid function characteristics (S_GATNE-T), attributes the multiplex heterogeneous network and introduces softsign and sigmoid function characteristics of attention mechanism and adds softsign function characteristics of the fully connected layer (SN_GATNE-T), which attributed multiplex heterogeneous networks and introduced softsign and sigmoid functions The characteristic attention mechanism and the fully connected layer with the softsign function characteristic and the Adam optimizer (SSN_GATNE-T) are used for optimization, and other parameter settings remain unchanged.

From the results of the various ablation experiments shown in Table 11, the attribute multiplexing heterogeneous network and the attention mechanism that introduces the softsign and sigmoid function characteristics, the fully connected layer that adds the

softsign function characteristics, and the optimization by the Adam optimizer (SSN_GATNE-T) have an important impact on the recommendation model. The evaluation index ROC-AUC of the model reached 0.8583 and 0.9754 on the YouTube and Amazon datasets, respectively; PR-AUC reached 0.8437 and 0.9717, respectively; F1 reached 0.7812 and 0.9386, respectively. We tested our model on the YouTube dataset and found that compared to only introducing attributed multiplex heterogeneous networks and the original attention mechanism (GATNE-T), our model can better compare the obtained nodes. The relevant information between edge-types is better processed and analysed, so the three evaluation indexes of ROC-AUC, PR-AUC and F1 increased by 1.44%, 2.98% and 1.68%, respectively. Compared to the attribute multiplexing heterogeneous network, we improved the introduction of the attention mechanism (S_GATNE-T) with softsign and sigmoid function characteristics. Our model can better reduce the information mined by the model through the interaction of different node attributes and different edge-types, which can be used for accurate recommendation. The potential information loss is considered, and the recommended evaluation indexes have been increased by 0.96%, 1.98%, and 1.43%, respectively, compared to the attribute multiplexing heterogeneous network and the introduction of the softsign and sigmoid function characteristics of the attention mechanism and the addition of the softsign function. The characteristic fully connected layer (SN_GATNE-T), our model can converge faster and better, and its recommended evaluation indexes have been increased by 0.55%, 0.97% and 0.46%, respectively. Through our SSN_GATNE-T model to test the recommendation effect on the Amazon data set, we found that our SSN_GATNE-T model has improved three evaluation indexes compared with other ablation models, but compared with the improvement of the evaluation index F1, the other two have not improved. The improvement effect of the evaluation index is not as obvious as the improvement of F1. Compared with the GATNE-T, S_GATNE-T and SN_GATNE-T models, the evaluation index F1 of the SSN_GATNE-T model increased by 1.07%, 1.16% and 0.46%, respectively. Compared with other models, our SSN_GATNE-T model can not only better process and analyse the related information between the obtained nodes and edge types but also reduce the number of users that can be mined for accurate recommendation. The loss of potential information, in addition, can make the model converge better. Through the attribute multiplexing heterogeneous network and our improved attention mechanism, we can obtain the user's potential interests and hobbies for the most suitable recommendations for the user, cooperate with the fully connected layer that introduces the softsign function to process the model, and then optimize the effectiveness of the recommendation model by the Adam optimizer for accurate recommendation effects.

To better describe the improvement in the recommendation performance of each module of our SSN_GATNE-T model in the YouTube and Amazon datasets, we made the experimental results of our model's ablation experiment on the two datasets into Figs. 10 and 11. The line charts of Fig. 10 show that the progress of each module of our model has greatly improved the recommendation performance of the YouTube dataset. Figure 11 shows that in the Amazon dataset recommendation process, although our model has

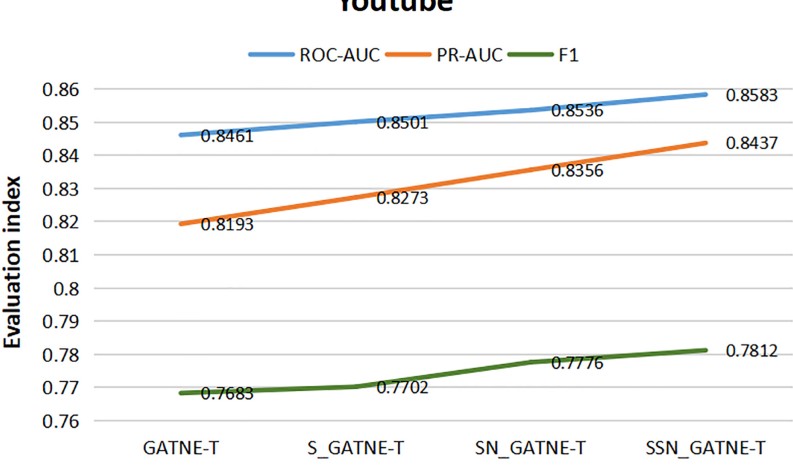

**Figure 10 YouTube data set ablation comparison line chart.**

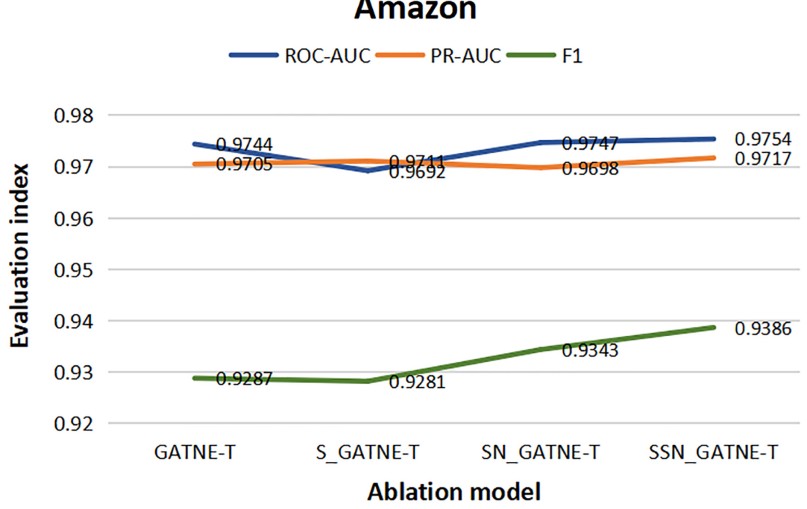

**Figure 11 Amazon data set ablation comparison line chart.**

improved the three evaluation indexes of ROC-AUC, PR-AUC and F1, the most obvious improvement effect is F1, and the improvement of each module has been improved very well, showing the effectiveness of our model in the recommendation system.

# CONCLUSION

In this article, the SSN_GATNE-T model that we propose first reuses heterogeneous networks through attributes that can not only help us deal with a single type of node network model but also help us better obtain tens of billions of nodes and many edges. The information of each attribute node in the user-item interaction contained in the large-scale network constructed by the type and the improved attention mechanism can help

annotation obtain more useful attribute node-related information. In the fully connected layer part of the model, we apply the characteristics of the softsign function, which can better reduce the users who are mined from the information of the user interaction between different nodes and edge-type attributes through the attributed multiplex heterogeneous network and the improved attention mechanism. The loss of potential information that can be used for accurate recommendations has achieved good results in mining user interest preferences. For the optimization of the model, we use the Adam optimizer, which can not only make our model converge faster but also greatly help the model's parameter adjustment. In addition, the SSN_GATNE-T model can better obtain the potential information between the node and edge attribute interaction in the recommendation system. Taking the YouTube dataset recommendation as an example, this network can better obtain the various attributes in the user-item interaction. The information of the node is to mine the potential information existing in the two nodes and the five edge types, and the improved attention mechanism contained in the inside can help mark the data to obtain more useful information. Because we have obtained more potential information from users, we have more information to provide users with more accurate recommendations. We conducted extensive experiments on the YouTube and Amazon datasets. The evaluation index ROC-AUC of the model reached 0.8583 and 0.9754 on the YouTube and Amazon datasets, respectively; PR-AUC reached 0.8437 and 0.9717, respectively; F1 reached 0.7812 and 0.9386, respectively. This experimental result proves that the SSN_GATNE-T model has significant advantages over the current mainstream recommendation algorithms, and this model can solve the cold start problem well.

## ACKNOWLEDGEMENTS

Thank you to Mr. Jinyong Cheng for his guidance and help in debugging the code and writing the thesis.

### Funding

This work was supported by the National Key R&D Program of China with grant No. 2019YFB2102600. The funders had no role in study design, data collection and analysis, decision to publish, or preparation of the manuscript.

### Grant Disclosures

The following grant information was disclosed by the authors:
National Key R&D Program of China: 2019YFB2102600.

### Competing Interests

The authors declare that they have no competing interests.

## Author Contributions

- Zhisheng Yang conceived and designed the experiments, performed the experiments, analyzed the data, performed the computation work, prepared figures and/or tables, authored or reviewed drafts of the paper, and approved the final draft.
- Jinyong Cheng conceived and designed the experiments, performed the experiments, analyzed the data, prepared figures and/or tables, authored or reviewed drafts of the paper, and approved the final draft.

## Data Availability

The datasets are available from the Amazon product data site: http://jmcauley.ucsd.edu/data/amazon and the YouTube social network and ground-truth communities site: https://snap.stanford.edu/data/com-Youtube.html.

## Supplemental Information

Supplemental information for this article can be found online at http://dx.doi.org/10.7717/peerj-cs.822#supplemental-information.

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
