# Peer review of "Recommendation algorithm based on attributed multiplex heterogeneous network"

_PeerJ Computer Science, doi:10.7717/peerj-cs.822_

## Round 0.1 · original submission · Major Revisions

We have received three consistent reports for the manuscript. Please revise the paper and provide a detailed response letter. Please note that I do not expect you to cite any recommended references unless essential. I hope to receive your revised paper. Thanks.

Reviewer 1 ·

Basic reporting

1. Novelty of idea is not clear. What is your main impact from the research and results? Make clear presentation also in relation to recent ideas in the field of deep learning.
2. Related models to present: MobileGCN applied to low-dimensional node feature learning, Self-attention negative feedback network for real-time image super-resolution.
3. Revise abstract to better show your main achievement and results.

Experimental design

1. Proposed in fig. 1 – fig. 2 model is not clear. What is the main input to this model? How do you preprocess info on the input?
2. What features are considered by your model? It is not presented how do you make them from the input. Do you use normalization before processing?
3. How is the model of graph in fig. 3 constructed?
4. Why Adam was used in your model? Did you test other algorithms?

Validity of the findings

1. It is not necessary to use probability in eq. (15) – eq. (16). Do you really use probability in your model? Can you show in which place of model it is used?
2. Your model needs comparisons to other model and comparisons on other data.
3. It is not clear what is your research hypothesis for amazon or youtube data. We read about results but what actually do you want to verify?

Additional comments

Images need better quality, since resolution is low and will have bad outlook in paper

Reviewer 2 ·

Basic reporting

The paper had proposed the recommendation algorithm based on attributed multiplex heterogeneous network. The attributed multiplex heterogeneous network can help obtain more user-item information with more attributes. The proposed framework SSN_GATNE-T was tested on two different types of datasets, Amazon and YouTube. The proposed algorithm improves the three evaluation indexes, especially the evaluation index F1. I suggest this paper be returned to authors for some related revisions, I hope those follow comments will help the authors to improve this paper.

Experimental design

I suggest the authors amend related description and depict with experiments.

Validity of the findings

1. The abstract and conclusion need to be improved. The abstract must be a concise yet comprehensive reflection of what is in your paper. Please modify the abstract according to “motivation, description, results and conclusion” parts. I suggest extending the conclusions section to focus on the results you get, the method you propose, and their significance.
2. What is the motivation of the proposed method? The details of motivation and innovations are important for potential readers and journals. Please add this detailed description in the last paragraph in section I. Please modify the paragraph according to "For this paper, the main contributions are as follows: (1) ......" to Section I. Please give the details of motivations.
3. The description of manuscript is very important for potential reader and other researchers. I encourage the authors to have their manuscript proof-edited by a native English speaker to enhance the level of paper presentation.
4. Please update references with recent paper in CVPR, ICCV, ECCV et al and Elsevier, Springer. In your section 1 and section 2, I suggest the authors amend several related literatures and corresponding references in recent years. For example: The improved image inpainting algorithm via encoder and similarity constraint (The Visual Computer); Research on image inpainting algorithm of improved total variation minimization method (Journal of Ambient Intelligence and Humanized Computing); The image annotation algorithm using convolutional features from intermediate layer of deep learning (Multimedia Tools and Applications); Image super-resolution reconstruction based on feature map attention mechanism (Applied Intelligence).
5. Please check all parameters in the manuscript and amend some related description of primary parameters. In section 3, please write the proposed algorithm in a proper algorithm/pseudocode format with Algorithm. Otherwise, it is very hard to follow. Some examples here: https://tex.stackexchange.com/questions/204592/how-to-format-a-pseudocode-algorithm
6. The section 2 is too short, and I suggest the authors amend the details of background and motivation. The main section of manuscript is section 3. I suggest the authors amend related depict of proposed method.

Additional comments

no

Reviewer 3 ·

Basic reporting

Recommender systems have been widely used in the real world. In addition to collaborative filtering, heterogeneous information networks (HINs) have attracted more attention in recent years. HIN-based recommendation algorithms have also been proved to be effective in a few practical recommendation scenarios. This work combines HINs and multiplex networks, a compound network made up of two or more sub-networks, to construct an attributed multiplex HIN for item recommendations. Besides, this work proposes an attention mechanism that can work well with the proposed network to recommend appropriate items for the target user. Basic reporting comments include:
1. This article is well organized. The presentation of this paper can be further improved with the help of native speakers of English.
2. All the figures of this article could be replaced with high-resolution ones.
3. In the Introduction section, the motivation of this work is not clear. Why did the work combine HINs and multiplex networks? The authors should point out the background, shortcomings of the existing models, and challenges.
4. A few key references of the related work were not included and analyzed in the second section. The authors should carefully review the studies on HIN-based embedding and multiplex network embedding and add those missing works in the revised manuscript.
[1] Fenfang Xie, Angyu Zheng, Liang Chen, Zibin Zheng: Attentive Meta-graph Embedding for item Recommendation in heterogeneous information networks. Knowl. Based Syst. 211: 106524 (2021)
[2] Léo Pio-Lopez, Alberto Valdeolivas, Laurent Tichit, Élisabeth Remy, Anaïs Baudot: MultiVERSE: a multiplex and multiplex-heterogeneous network embedding approach. Sci. Rep. 11: 8794 (2021)
[3] Qiwei Zhong, Yang Liu, Xiang Ao, Binbin Hu, Jinghua Feng, Jiayu Tang, Qing He: Financial Defaulter Detection on Online Credit Payment via Multi-view Attributed Heterogeneous Information Network. WWW 2020: 785-795
[4] Zhijun Liu, Chao Huang, Yanwei Yu, Baode Fan, Junyu Dong: Fast Attributed Multiplex Heterogeneous Network Embedding. CIKM 2020: 995-1004
[5] Yukuo Cen, Xu Zou, Jianwei Zhang, Hongxia Yang, Jingren Zhou, Jie Tang: Representation Learning for Attributed Multiplex Heterogeneous Network. KDD 2019: 1358-1368
[6] Binbin Hu, Zhiqiang Zhang, Chuan Shi, Jun Zhou, Xiaolong Li, Yuan Qi: Cash-Out User Detection Based on Attributed Heterogeneous Information Network with a Hierarchical Attention Mechanism. AAAI 2019: 946-953
[7] Yiming Zhang, Yujie Fan, Yanfang Ye, Liang Zhao, Chuan Shi: Key Player Identification in Underground Forums over Attributed Heterogeneous Information Network Embedding Framework. CIKM 2019: 549-558
5. The concept of multiplex networks has been formally defined in the field of network science. However, the authors did not formulate it in this article. How is the attention mechanism applied to the proposed network that is essentially a multiplex network?

Experimental design

This study experimented on two real-world datasets and demonstrated the effectiveness of the proposed approach in terms of three evaluation metrics. The reviewer’s concerns on experimental design include:
1. The selected baselines have been proposed for three years. The authors should add more recent approaches (e.g., the above refs. [2], [4], and [5]) based on attributed multiplex heterogeneous networks to compare with the proposed approach.
2. In addition to the parameter setting of the proposed approach, the authors should introduce the settings of each baseline’s hyperparameters, which may impact the performance of each baseline.
3. The implementation code of the proposed approach is not available for research on the Internet.
4. In addition to the ablation experiment, the authors should discuss the impacts of some critical hyperparameters (e.g., the dimension of embeddings and the layer number of a multiplex network) on model performance.

Validity of the findings

1. The motivation of the Fridman test is not clear. For example, what are the actual meanings of node1 and node2? Why did the Fridman test on the two datasets obtain two different results, i.e., the original hypothesis was retained on the Youtube dataset, while the original hypothesis was rejected on the Amazon dataset.
2. If possible, the authors should add a subsection to discuss some potential threats to the validity of this work, including construct validity, internal validity, and external validity.

Additional comments

No comments

---

## Round 0.2 · accepted · Accept

The paper can be accepted. Congratulations!

Reviewer 1 ·

Basic reporting

i think paper is revised

Experimental design

i think paper is revised

Validity of the findings

i think paper is revised

Additional comments

i think paper is revised